# PD-L1 Expression by RNA-Sequencing in Non-Small Cell Lung Cancer: Concordance with Immunohistochemistry and Associations with Pembrolizumab Treatment Outcomes

**DOI:** 10.3390/cancers15194789

**Published:** 2023-09-29

**Authors:** Mary K. Nesline, Rebecca A. Previs, Grace K. Dy, Lei Deng, Yong Hee Lee, Paul DePietro, Shengle Zhang, Nathan Meyers, Eric Severson, Shakti Ramkissoon, Sarabjot Pabla, Jeffrey M. Conroy

**Affiliations:** 1Labcorp Oncology, Durham, NC 27560, USA; rebeccaann.previs@covance.com (R.A.P.); eric.severson@labcorp.com (E.S.); shakti.ramkissoon@labcorp.com (S.R.); 2Division of Gynecologic Oncology, Duke Cancer Institute, Durham, NC 27710, USA; 3Division of Medicine, Roswell Park Comprehensive Cancer Center, Buffalo, NY 14263, USA; grace.dy@roswellpark.org; 4Fred Hutchinson Cancer Center, Seattle, WA 98109, USA; lei.deng@roswellpark.org; 5Mantech International, Virginia Beach, VA 23452, USA; 6OmniSeq, Inc., Buffalo, NY 14203, USA; depietp@labcorp.com (P.D.); zhangs7@labcorp.com (S.Z.); meyersn@labcorp.com (N.M.); pablas@labcorp.com (S.P.); conroj2@labcorp.com (J.M.C.); 7Wake Forest Baptist Comprehensive Cancer Center, Winston-Salem, NC 27157, USA

**Keywords:** PD-L1, immunohistochemistry, RNA-seq, non-small cell lung cancer, immunotherapy

## Abstract

**Simple Summary:**

We compared RNA next-generation sequencing (RNA-seq) to standard immunohistochemistry (IHC) for PD-L1 expression measurement and associations with pembrolizumab immunotherapy outcomes in NSCLC patient tumors. RNA-seq and IHC PD-L1 score interpretation agreed for 80% of patients, and an RNA-seq “high” cutoff that accurately separated IHC high versus low or negative expression was identified. However, RNA-seq could not discern PD-L1 IHC from negative expression due to the limited sensitivity of IHC as a reference test. High PD-L1 expression by RNA-seq alone and in combination with IHC high or low status was associated with better pembrolizumab outcomes in NSCLC patients than IHC alone.

**Abstract:**

Programmed cell death ligand (PD-L1) expression by immunohistochemistry (IHC) lacks sensitivity for pembrolizumab immunotherapy selection in non-small cell lung cancer (NSCLC), particularly for tumors with low expression. We retrospectively evaluated transcriptomic PD-L1 by mRNA next-generation sequencing (RNA-seq). In an unselected NSCLC patient cohort (n = 3168) tested during standard care (2017–2021), PD-L1 IHC and RNA-seq demonstrated moderate concordance, with 80% agreement overall. Most discordant cases were either low or negative for PD-L1 expression by IHC but high by RNA-seq. RNA-seq accurately discriminated PD-L1 IHC high from low tumors by receiver operator curve (ROC) analysis but could not distinguish PD-L1 IHC low from negative tumors. In a separate pembrolizumab monotherapy cohort (n = 102), NSCLC tumors classified as PD-L1 high versus not high by RNA-seq had significantly improved response, progression-free survival, and overall survival as an individual measure and in combination with IHC high or low status. PD-L1 IHC status (high or low) trended toward but had no significant associations with improved outcomes. Conventional PD-L1 IHC testing has inherent limitations, making it an imperfect reference standard for evaluating novel testing technologies. RNA-seq offers an objective PD-L1 measure that could represent a complementary method to IHC to improve NSCLC patient selection for immunotherapy.

## 1. Introduction

Immune checkpoint inhibitors have improved outcomes over chemotherapy for patients with non-small cell lung cancer (NSCLC) [1,2]. Checkpoint inhibitors are a class of immunotherapy drugs designed to enhance the body’s natural immune response against cancer cells. They work by targeting specific proteins, known as immune checkpoints, that regulate the immune system’s response to prevent it from attacking normal cells. When these checkpoints are inhibited, it allows the immune system to recognize and attack cancer cells more effectively. One of the most well-known immune checkpoints is the programmed cell death protein 1 (PD-1) and its ligand, programmed death-ligand 1 (PD-L1). When PD-1 on the surface of T-cells binds with PD-L1 on the surface of cancer cells, it sends a signal that suppresses the immune response, allowing cancer cells to evade detection and destruction by the immune system and leading to PD-L1 protein expression [3,4,5]. Currently, eligible NSCLC patients receive pembrolizumab anti-PD-1-containing regimens as a new frontline treatment standard based on the expression of PD-L1 [6].

The immunohistochemistry (IHC) 22C3 FDA-approved companion diagnostic assay for evaluation of PD-L1 protein expression in NSCLC tissue biopsy specimens is indicated for selecting patients for pembrolizumab immunotherapy [7]. While IHC 22C3 measurement technology is intended to capture the full range of possible PD-L1 expression across tumors, the test lacks high sensitivity, misclassifies tumors, and may negatively impact therapeutic efficacy [8]. Problems with the IHC 22C3 assay arise from challenges to pathologist interpretation from limited NSCLC core biopsy specimens; PD-L1 expression can be highly heterogeneous within a given tumor, between primary and metastatic sites, and for different specimen types [9]. This introduces complexity to calculating an accurate tumor proportion score (TPS), which is the percentage of tumor cells subjectively showing complete or partial linear membranous staining, in accordance with FDA-approved pathologist scoring guidelines for IHC 22C3 testing [10,11,12,13,14].

The IHC 22C3 assay also renders an incomplete picture of PD-L1 in NSCLC because it excludes counting expression on immune cells, which are regulated by a different mechanism with an independent role in immunotherapy response. High tumoral expression of PD-L1 reflects epigenetic dysregulation of the PD-L1 gene and is associated with poor immune cell infiltration, desmoplastic stroma, and mesenchymal histologic features. In contrast, high immune cell expression of PD-L1 is more common and reflects IFN-γ-induced adaptive regulation, accompanied by increased tumor-infiltrating lymphocytes and effector T-cells [15]. Capturing PD-L1 expression on both tumor and immune cells in the landmark KEYNOTE-001 pembrolizumab trial using IHC was initially evaluated based on the 22C3 assay prototype used to enroll patients. The trial showed PD-L1 IHC 22C3 expression in at least 50% of tumor cells was correlated with improved response, which led to the initial FDA approval of pembrolizumab monotherapy in the NSCLC second-line treatment setting. Post-trial, the PD-L1 IHC 22C3 assay was modified to exclude immune cell staining from 22C3 companion diagnostic scoring validation. PD-L1 TPS ≥ 50% was selected as the IHC 22C3 companion diagnostic cutoff from among highly similar results for four scoring methods because it was the simplest approach to implement in clinical practice, with the investigators concluding PD-L1 expression in NSCLC is tumor-specific [16,17].

Since KEYNOTE-001, however, the 50% TPS cutoff for high PD-L1 expression by IHC 22C3 has proved problematic. Most notably, the KEYNOTE-042 trial initially showed NSCLC patients with any positive PD-L1 IHC staining (≥1% TPS) had a survival advantage for frontline pembrolizumab over chemotherapy (16.7 vs. 12.1 months), followed by the FDA’s lowering of the cutoff from ≥50% TPS to ≥1% TPS on the label indication. However, a later pre-specified subgroup analysis showed KEYNOTE-042 patients with low PD-L1 IHC expression had no significant survival benefit compared with patients who received chemotherapy (13.4 vs. 12.1 months) [18,19,20,21]. As a result, professional practice guidelines now recommend NSCLC patients with high (≥50% TPS) PD-L1 IHC expression receive frontline pembrolizumab monotherapy, while patients with low (≥1% TPS) or negative (<1% TPS) PD-L1 IHC expression receive combination pembrolizumab plus chemotherapy, in contradiction to the FDA label indication [22].

The PD-L1 IHC SP142 companion diagnostic assay for atezolizumab in NSCLC, which does include tumor and immune cell staining [23,24], has also struggled to demonstrate concordance and predictive value. A comparison of the Ventana SP142, Ventana SP263, and Dako 22C3 IHC-validated assays showed the percentage of PD-L1-positive cells was consistently higher among immune cells than tumor cells across all three tests, but assay concordance was lower for immune cell vs. tumoral expression measurement [25]. Other studies have also shown both greater variability, subjectivity, and poor inter-pathologist agreement in scoring PD-L1 expression on immune cells compared with tumor cells when using the percentage of tumor with immune cell infiltrate [14,26,27]. Despite these inconsistencies, PD-L1 expression on either tumor or immune cells has been associated with an improved response to checkpoint blockade. In the POPLAR trial, second-line, NSCLC patients with PD-L1 IHC SP142 expression ≥ 1% in either tumor or immune cells had significantly better overall survival when treated with atezolizumab monotherapy versus docetaxel (median 15.5 months versus 9.2 months, *p* = 0.005) [28]. Problematically, however, the later IMpower110 NSCLC frontline atezolizumab trial compared IHC SP142 to IHC 22C3 in pre-specified exploratory analyses and showed the SP142 assay was less sensitive for staining both tumor and immune cell PD-L1 [29].

Continued advancement in precision oncology clinical care requires a re-evaluation of testing methods for established biomarkers, especially when accumulating data demonstrates the imperfect nature of existing approaches like IHC. Compared with IHC, next-generation sequencing (NGS) objectively measures PD-L1 mRNA (RNA-seq) by digitally counting target molecules, enabling more sensitive and precise transcriptome quantification and a continuum of measurements across a large dynamic range of expression. RNA-seq methods have also previously been shown to be positively correlated with IHC testing for PD-L1 [25,30,31,32,33].

IHC and RNA-seq are distinct techniques with their own advantages and limitations. RNA-seq is more sensitive; however, standard NGS does not discern the detection of PD-L1 mRNA in tumor versus stroma. RNA-seq does not quantitatively assign PD-L1 expression levels to specific cell types, although the clinical value of making this distinction using current IHC techniques also appears to be variable, with NSCLC tumors that harbor one and not the other potentially representing different patient subgroups [15,25,34]. In addition to these divergent detection methods, underlying biological mechanisms can also result in differences in PD-L1 expression, including mRNA stability, which can affect the availability of translation into proteins, translational regulation factors, post-translational modifications, and protein degradation. Understanding these mechanisms is also crucial for the accurate interpretation of PD-L1 expression data.

In the absence of a single, perfect technology and with the growing use of NGS in NSCLC patient standard care, we posit there may be a role for measuring PD-L1 expression by both IHC and RNA-seq. Here, we sought to address unanswered questions about the sensitivity, potential clinical utility, and real-world application of RNA-seq to improve the selection of NSCLC patients for immunotherapy.

## 2. Methods

### 2.1. Patient Cohorts

Approval for this study was obtained from an independent institutional review board (IRB), the Western Copernicus Group (www.wcgclinical.com (accessed on 14 September 2023)), under protocol #1340120. NSCLC patients treated between 2017 and 2021 at over 300 oncology practices in the United States whose FFPE tumor tissue biopsy samples underwent testing in a commercial laboratory by clinically validated comprehensive immune profiling (CIP) were considered for inclusion. Cases with non-NSCLC histology failed RNA-seq testing, or failed/inconclusive IHC results were excluded. The final set of NSCLC patients (n = 3270) was divided into 2 groups—an unselected cohort with unknown treatment status (n = 3168) to evaluate assay concordance and explore alternative RNA-seq cut-offs, and a cohort of pembrolizumab monotherapy-treated patients (n = 102) to evaluate outcomes associations with results for each assay. The pembrolizumab outcomes cohort included patients with stage IIIB/IV disease who were confirmed negative for ALK and EGFR mutations, received at least one dose of pembrolizumab monotherapy post-CIP test and had at least 90 days of follow-up (excluding death) post-first dose.

### 2.2. PD-L1 Expression Testing and Clinical Reporting Cutoffs for IHC and RNA-Seq

PD-L1 expression by IHC and RNA-seq was assessed for the same tissue from each patient as part of a larger comprehensive immune profiling (CIP) assay ordered alongside comprehensive genomic profiling (CGP) as part of standard patient care. For CIP, formalin-fixed paraffin-embedded (FFPE) tissue specimens (3–5 unstained positively charged slides cut at 5 µm), plus one hematoxylin and eosin-stained slide, were submitted along with the corresponding pathology report for each order. Histology confirmation, tumor enrichment, and estimation of the minimum required 20% tumor purity for RNA-seq were determined through microscopic review and cell selection by a board-certified pathologist. PD-L1 IHC protein expression was measured with the FDA-approved Dako PD-L1 IHC 22C3 pharmDx assay (Agilent, Santa Clara, CA) on Autostainer Link 48 (Agilent, Santa Clara, CA) and scored as the percentage of viable tumor cells showing % membrane staining at any intensity as a tumor proportion score (TPS) following FDA scoring guidelines [7]. TPS scores were interpreted as ≥50% (high), 1–49% (low), or <1% (negative) per NCCN guideline recommendations for pembrolizumab regimen treatment selection [22].

PD-L1 by RNA-seq was semi-quantitatively measured by analytically validated amplicon-based targeted RNA next-generation sequencing as part of a 64-transcript digital gene expression panel relating to the anticancer immune response as previously described [35]. RNA was extracted from FFPE using the truXTRAC FFPE extraction kit (Covaris, Inc., Woburn, MA, USA), with some modifications to the instructions by the manufacturer. After purification, RNA was dissolved in 50 µL of water, and the yield was measured through the Quant-iT RNA HS assay (Thermo Fisher Scientific, Waltham, MA, USA), as per the manufacturer’s recommendation. For library preparation, the pre-defined titer of 10 ng RNA was referred to as the acceptance criterion. Torrent Suite’s plugin immuneResponseRNA (v5.2.0.0) was used for the absolute reading of the RNA sequence. The RNA expression of 64 different genes was measured. Transcript abundance was normalized to internal housekeeping gene profiles and ranked (0–100 percentile) in a standardized manner to a reference population of 735 tumors spanning 35 histologies as previously described [35]. In clinical testing, a percentile rank ≥75 is interpreted and reported as the cutoff for “high” versus “not high” expression for each of the 64 RNA-seq genes in the CIP panel, including PD-L1. The clinical cutoff for reporting was determined based on the rank-score distributions observed in assay validation. RNA-seq expression results for immune genes that are drug targets in clinical trials are currently reported for research use only.

### 2.3. Clinical Data and Pembrolizumab Outcomes Measures

For all patients, clinicopathological data included age, sex, histology (squamous or non-squamous), specimen type (primary or metastatic), and anatomic site of metastasis (adrenal gland, bone, brain, liver, lymph node, pleura, soft tissue, and others). Additional clinical data for pembrolizumab-treated patients included smoking history (ever/never), pembrolizumab line of therapy (first, second, subsequent), weighted performance status, calculated from Eastern Cooperative Group (ECOG) scores (0–5) ascertained at post-test patient encounters between date of CIP report until date of last dose of pembrolizumab or death, response status (complete, partial, stable, or progressive disease), progression-free survival (PFS), calculated as the number of months between first dose of pembrolizumab and progression date, and overall survival (OS), calculated as the number of months between the first dose of pembrolizumab and date of death.

### 2.4. Statistical Analysis

PD-L1 assay concordance between IHC and RNA-seq was assessed by independent samples Kruskal-Wallis median test and proportion analysis (chi-square) using clinical cutoffs for IHC tumor proportion score (TPS) as high (≥50%), low (1–49%), or negative (<1%), or for RNA-seq percentile rank as high (≥75) or not high (<75). Receiver operator characteristic (ROC) analysis was also used in the concordance analysis cohort to explore the accuracy (sensitivity and specificity) of potential alternative cutoffs for RNA-seq using IHC as the gold reference standard.

In the pembrolizumab-treated cohort, we assessed the predictive value of individual and combined PD-L1 marker status by RNA-seq and IHC for patient treatment outcomes. Proportion analysis (chi-square) was used to assess for differences in treatment response by marker status, and Kaplan-Meier analysis assessed associations between marker status and survival. Cox proportional hazards multiple regression models were used to evaluate the predictive value of PD-L1 by marker status for PFS and OS, adjusting for potential covariates, including age, sex, performance status (weighted ECOG), histology, smoking history, and pembrolizumab line of treatment. All data were analyzed using SPSS Statistics for Windows, version 26.0 (IBM Corp., Armonk, NY, USA).

## 3. Results

### 3.1. Concordance between IHC 22C3 and RNA-Seq for PD-L1 Expression in NSCLC

Although IHC is scored on a 0–100 scale for TPS, results were significantly left-skewed toward cases with low (1–49% TPS) expression, with the largest proportion of cases (37.7%) also falling in this group (Figure 1A). Most cases were also classified as “not high” by RNA-seq (n = 2036, 64.3%), falling below the ≥75 percentile rank cutoff used in clinical reporting (Figure 1B). However, a greater number of cases were high by RNA-seq compared with IHC (35.7% vs. 30.8%, respectively).

Overall, IHC and RNA-seq demonstrated moderate concordance for PD-L1 expression. As seen in Figure 1C,D, the medians and distributions of IHC TPS results by RNA-seq percentile rank clinical cutoff group (high or not high) were significantly different from one another, as were the medians and distributions of RNA-seq percentile rank results by IHC TPS clinical cutoff group (high, low, or negative) (Kruskal-Wallis test with significance values adjusted by Bonferroni correction for multiple tests; *p* < 0.001).

Furthermore, as seen in Figure 1E, 2539 (80%) of PD-L1 IHC and RNA-seq results were concordant with the clinical cutoff group. The proportion of NSCLC patients in each IHC cutoff group (high, low, or negative) was significantly associated with the proportion of patients in the corresponding RNA-seq cutoff group (high or not high), as was the reverse scenario (Chi-Square < 0.001 for each scenario). Most tumors (n = 739, 76%) that were high by IHC were also high by RNA-seq. Similarly, most tumors that were low (n = 925, 78%) or negative (n = 875, 88%) by IHC were also not high by RNA-seq.

Approximately 20% (629/3168) of all NSCLC patient tumors in the cohort had discordant findings for PD-L1 expression by IHC and RNA-seq. The largest number of discordant cases had either low (268) or negative (125) tumoral expression by IHC but were determined to be high expressers by RNA-seq. The remaining discordant cases (236) were high by IHC but not high by RNA-seq (Figure 1E). Discordant PD-L1 expression between IHC and RNA-seq is also not readily discernable under the microscope, as illustrated by example photomicrographs (Figure 1F). 

### 3.2. Alternative PD-L1 RNA-Seq Cutoffs and Assay Performance

Given the known suboptimal IHC test sensitivity for PD-L1, particularly at low levels of expression, we explored alternative PD-L1 RNA-seq cutoffs that might better discern IHC results as the standard reference test than the 75th percentile rank historically used in clinical reporting. Receiver operator characteristic (ROC) analyses were performed using five sets of NSCLC patient tumors with two different PD-L1 IHC TPS cutoff groups in each set: IHC high versus negative (Figure 2A); IHC low versus negative (Figure 2B); IHC high versus low (Figure 2C); IHC high + low versus negative (Figure 2D); and IHC high (TPS ≥ 50) versus low + negative (Figure 2E). A summary Youden index score (sensitivity + 1-specificity) was calculated for each ROC curve, with an index score ≥ 50% considered the empirical benchmark of an efficacious diagnostic test [36].

RNA-seq was best able to differentiate IHC-high versus negative tumors at a 69th percentile rank cutoff for PD-L1, demonstrating high sensitivity and specificity (Figure 2A). However, a more pressing and clinically relevant problem is that oncologists need to differentiate between IHC-high versus low tumors to support the selection of pembrolizumab monotherapy versus combination pembrolizumab plus chemotherapy. For this scenario, RNA-seq accurately classified patients in each group at a 76th percentile rank cutoff (AUC = 0.84, Youden index = 53; *p* < 0.0001) (Figure 2B). Similarly, RNA-seq accurately classified IHC-high versus low + negative tumors (AUC = 0.87, Youden index = 58; *p* < 0.0001) (Figure 2C). RNA-seq, however, could not discern IHC low versus negative tumors (Figure 2D) or IHC high + low versus negative tumors (Figure 2E), lacking sensitivity and specificity for both scenarios. Given the high sensitivity and specificity of a 76th percentile RNA-seq rank cutoff identified in ROC analysis for discerning the most clinically relevant group of PD-L1 expression cases (IHC high versus low) and its proximity to the 75th percentile rank cutoff already in use for clinically reporting a high result, subsequent analyses in the study were based on a 75th percentile rank cutoff for PD-L1 by RNA-seq.

### 3.3. Associations with Pembrolizumab Monotherapy Outcomes

The potential predictive value of PD-L1 by RNA-seq for pembrolizumab monotherapy response, progression-free survival (PFS), and overall survival (OS) was explored using the 75th percentile rank cutoff in a cohort of NSCLC patients who received pembrolizumab monotherapy as their first immunotherapy following testing (n = 102) (Table 1). Given recommendations from professional practice guidelines to select NSCLC patients with high PD-L1 expression for pembrolizumab, this real-world treatment cohort was biased for PD-L1 high-expressing tumors (82.3% of patients). These patients were also more likely to have a smoking history (89.2%), non-squamous histology (81.4%), a weighted ECOG performance status between 1 and 2 (61.8%), and have received their first course of pembrolizumab in the frontline setting (84.3%). Very few patients in the pembrolizumab-treated cohort received prior (non-EGFR/ALK) targeted therapy (2.9%) or immunotherapy (2%).

Of 102 pembrolizumab-treated patients, 74 (72.5%) had required scan data and measurable disease at baseline and were retrospectively evaluable for pembrolizumab response based on RECIST v1.1.

As shown in Figure 3, as an individual measure, patients with high PD-L1 expression by RNA-seq had a significantly better response rate (RR) for pembrolizumab than those whose tumors were below the 75th percentile rank cutoff for high expression (65% vs. 18%, *p* < 0.001). In contrast, we observed no significant differences in response rates for pembrolizumab monotherapy between IHC high (≥50% TPS) versus low (<1% TPS) expression (54% vs. 38%, *p* = 0.306).

Patients with combined PD-L1 “double high” status by both RNA-seq and IHC had the best response rate of any group (66%). Patients with “double low” (IHC low + RNA not high) status fared better than patients who were high by IHC but not high by RNA-seq (25% vs. 14%, respectively).

Comparing response rates for combined versus single marker status, patients that were high by RNA-seq + low by IHC had better response rates than patients who were high by IHC alone (60% vs. 54%). Patients who were low by IHC and not high by RNA-seq had worse response rates than patients who were low by IHC alone (25% vs. 38%).

Kaplan-Meier analysis was used to evaluate associations between PD-L1 marker status for IHC and RNA-seq with median progression-free survival (PFS) and overall survival (OS) for pembrolizumab-treated NSCLC patients in the validation cohort. There were no significant differences between median PFS or OS for patients with PD-L1 IHC high (≥50% TPS) versus low (1–49% TPS) tumors (Figure 4A,B). In contrast, patients with RNA-seq high (≥75 rank) versus not high (<75 rank) expression had significantly improved median PFS (19.2 versus 4.0 months, *p* < 0.001) (Figure 4C) and median OS (24.3 vs. 5.4 months, *p* = 0.035) (Figure 4D).

For combined PD-L1 marker status, patients with PD-L1 RNA-high tumors had significantly better PFS than patients with tumors that were not high by PD-L1 RNA, regardless of PD-L1 IHC status (Figure 4E). Specifically, patients with PD-L1 “double high” results had the best PFS (18 months), while patients who were high by RNA-seq and low by IHC had the next best PFS (14.1 months). However, patients with RNA-seq results that were not high had significantly worse PFS, even when IHC results were high (3.2 months) and low positive (5.7 months). Combined PD-L1 marker status did not achieve statistical significance for OS; however, results followed the same trend as PFS, with RNA-seq high vs. not high status demonstrating better survival, irrespective of PD-L1 IHC results (Figure 4F).

Cox proportional hazards regression models were used to assess the predictive value of IHC and RNA-seq PD-L1 marker status for pembrolizumab PFS and OS, adjusting for potential covariates—age, sex, histologic subtype (squamous vs. non-squamous), smoking history, ECOG performance status, and line of treatment (Figure 5). Performance status was a significant covariate (*p* < 0.001) in all multiple regression analyses. As an individual measure, only RNA-seq high (compared with not high) status was significantly associated with a reduction in the likelihood of death for both PFS (HR = 0.62; 95% CI = 0.42–0.82, *p* < 0.001) and OS (HR = 0.45; 95% CI = 0.24–0.84, *p* = 0.05).

For combined marker status, patients with PD-L1 “double high” results had the most significant reduction in risk of death for PFS compared with IHC low + RNA-seq not high status (HR = 0.32; CI = 0.14–0.77, *p* = 0.01). Combined marker status was not significant for OS; however, results were directionally aligned with PD-L1 RNA marker status; when RNA results were high, patients had better HR for OS, and when RNA was not high, patients had worse HR for OS, regardless of IHC status.

## 4. Discussion

Higher levels of PD-L1 expression have been correlated with increased clinical benefit from immunotherapy compared with chemotherapy, but current IHC methods for PD-L1 measurement are imprecise for predictive use in NSCLC. Intratumoral heterogeneity, error-prone, subjective scoring methods, and limited tissue samples present challenges for reliable quantification of PD-L1 by IHC in NSCLC and lead to a lack of clinical test sensitivity [8,9,37,38]. These challenges are of particular concern to treating oncologists, as most NSCLC patient tumors do not have clear negative (<1% TPS) or high (≥50% TPS) PD-L1 expression but rather harbor low expression (1–49%). PD-L1 low expression is the subgroup that is also likely to be misclassified for PD-L1 status by IHC, contributing to unclear clinical efficacy for these patients [13,39].

Given the problems with current IHC testing, our study sought to evaluate the potential clinical utility of transcriptomic PD-L1 in a cohort of NSCLC patient tumors simultaneously tested by standard IHC and by RNA-seq. RNA-seq demonstrated reasonable overall concordance with IHC for PD-L1 scores and their interpretations using standard clinical reporting cutoffs for each test (Figure 1C,D), with 80% of cases demonstrating concordant results (Figure 1E). Using ROC analysis, our data also showed PD-L1 by RNA-seq accurately discerns IHC high (≥50% TPS) from IHC low (1–49% TPS) cases at a 76th percentile rank score (Figure 2B,C), a cutoff that is reasonably equivalent to the 75th percentile rank score already used in RNA-seq clinical reporting.

Our intent was to use IHC as the gold standard to define cutoffs for high, moderate, and low PD-L1 expression by RNA-seq. Problematically, however, we observed a significantly skewed distribution of TPS scores among PD-L1 IHC-low tumors toward 1% in a possible 1–49% range (Figure 1A). This reflects IHC 22C3 companion diagnostic scoring guidelines for pathologists, which call for calculation of the percentage of viable tumor cells showing linear membranous staining of *any* intensity and may ultimately drive over-representation of patients with very low PD-L1 expression by IHC, as well as some of the observed discord between IHC and RNA-seq in our study. As would be expected, a high proportion (78%) of PD-L1 IHC-low cases were also classified as not high for PD-L1 by RNA-seq (Figure 1E). Thus, our attempts to identify a cutoff within the RNA-seq “not high” group that would accurately discern IHC low (1–49% TPS) from IHC negative (<1% TPS) tumors failed (Figure 2D,E). This underscores the problems arising from the compulsory use of IHC as a reference standard and ultimately limits our evaluation of the clinical utility of RNA-seq as an individual measure beyond “high” versus “not high” status.

As individual measures, PD-L1 RNA-seq high versus low status was associated with improved pembrolizumab outcomes for all measures—response, progression-free survival (PFS), and overall survival (OS). We found similar trends but no significant associations with outcomes for PD-L1 IHC high or low versus negative status. Perhaps of greatest import, the evaluation of outcome associations for combined marker status demonstrated clear potential in every scenario for RNA-seq to further stratify tumors with high or low PD-L1 expression by IHC that are either more or less likely to respond and survive. Given that it has been shown in prospective clinical trials that PD-L1 expression by IHC in either tumor or immune cells is predictive of response to checkpoint blockade in NSCLC [28,40,41], our RNA-seq findings may be evidence that it is unnecessary to discern between tumor and stroma and that better assay sensitivity by NGS is more critical.

Of note, we found that NSCLC tumors for patients that were simultaneously high for PD-L1 expression by IHC but not by RNA-seq had a worse response rate (14%) than those with low expression by IHC alone (38%) or with “double low” expression by IHC and RNA-seq (25%). Even when PD-L1 is expressed, the absence of tumor-infiltrating lymphocytes in the tumor microenvironment, required to attack cancer cells, will limit pembrolizumab efficacy in these “cold” tumors. Another potential explanation in these cases, however, may be that IHC detected high levels of stable but non-functional PD-L1 protein and that IHC does not perfectly correlate with PD-L1′s functional activity [42].

There are also several limitations to our outcome analyses. Increasing PD-L1 IHC expression trended toward improved NSCLC pembrolizumab response and survival; however, our statistically negative results deviate from positive clinical trial findings and are to be viewed with caution. First, our real-world patient data are biased toward high PD-L1 IHC expression (82.3% of patients), as these patients are the most often treated by pembrolizumab monotherapy. In turn, a limited number of patients in our analysis harbored low PD-L1 expression by IHC (n = 18; 17.7%), challenging statistical comparisons. This stands in contrast to the number and proportion of patients with RNA-seq high (n = 69; 67.7%) and not high (n = 33, 32.3%) status, which is not as disparate. Unlike prospective trials, our real-world retrospective study was not powered to evaluate outcomes by assay technology for different levels of PD-L1 expression. This consideration is particularly relevant to the strength of our combined marker status findings for outcome associations.

## 5. Conclusions and Future Directions

Our findings support additional study of RNA-seq as a complementary testing strategy for standard IHC for PD-L1 expression. Combined measurement of PD-L1 by IHC and RNA-seq may better discern patients who are more or less likely to benefit from checkpoint inhibition, a particular problem for the largest subgroup of NSCLC patients with low PD-L1 expression by IHC and the least certainty of response to checkpoint inhibition. While it is possible that the development of new, more sensitive IHC technology can improve upon the current state of IHC testing for PD-L1, all biomarkers are susceptible to detection method bias. Furthermore, while it’s true that standard RNA sequencing does not inherently discriminate between tumor and stromal cells, advanced computational methods can help deconvolute the data to estimate the contribution of different cell types to the overall expression. These methods use known gene expression signatures associated with cell type-specific markers to infer their presence in a sample. This can provide insights into the contribution of stromal cells to PD-L1 expression.

Perhaps more importantly, growing evidence substantiates that the use of any one single marker for immunotherapy selection is a suboptimal strategy. Determining the likelihood of immunotherapy response requires measuring many more anti-cancer immune cycle responses and resistance markers in the tumor microenvironment, including those related to neoantigens, antigen presentation, DNA repair, oncogenic pathways, and their mediators [43]. Broad, comprehensive immune profiling by RNA-seq that includes PD-L1 in addition to expression for many other immune genes can be run in parallel with comprehensive genomic profiling to measure the growing spectrum of biomarkers needed for treatment selection in NSCLC. For example, in our own work, we have integrated tumor immunogenic and cell proliferation signatures with PD-L1 IHC and TMB testing and have shown that tumors that are at least moderately immunogenic have significantly better survival for checkpoint inhibition compared with highly proliferative tumors with weak immunogenicity. These signatures also identified patients who may benefit from checkpoint inhibition but were unlikely to respond based on standard PD-L1 IHC and TMB marker status alone [44]. Other investigations have also contributed new knowledge toward evolving the future use of RNA-seq signatures and multi-marker approaches to better assess responses to immunotherapy [45,46,47,48].

Advancing multi-marker approaches toward standard clinical use requires the inclusion of comprehensive immune profiling in early-stage diagnostic and drug development strategies using NGS-based technologies to overcome the technical challenges of IHC-based methods.

## Figures and Tables

**Figure 1 cancers-15-04789-f001:**
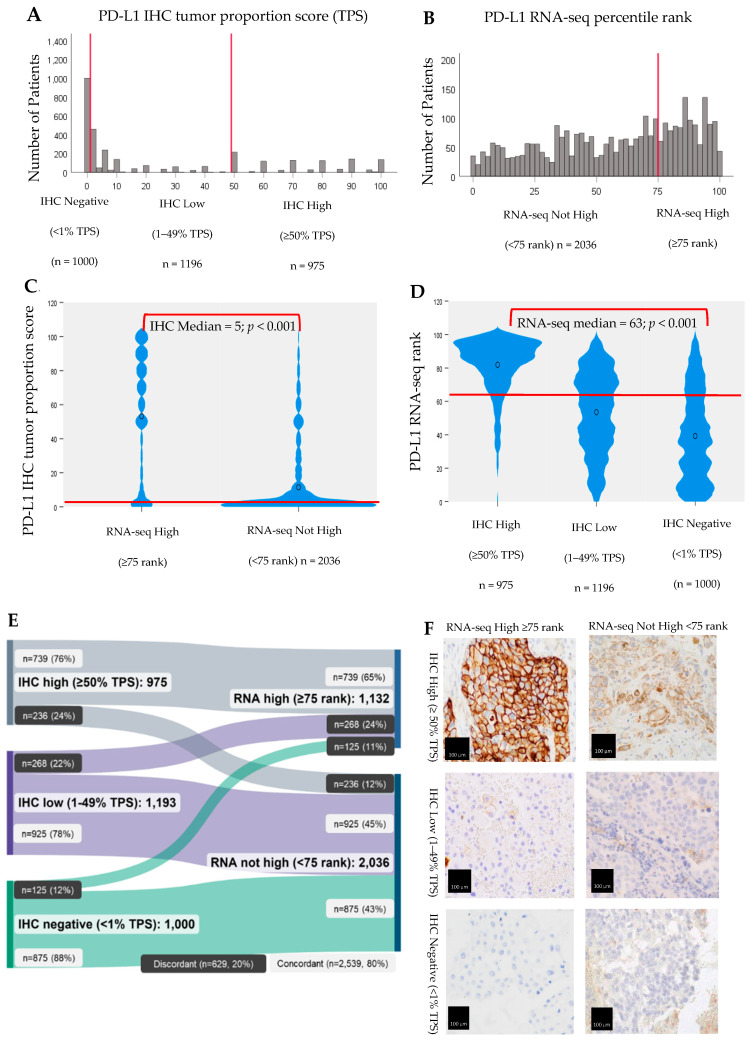
Concordance of IHC and RNA-seq for PD-L1 expression, NSCLC (n = 3168). (**A**,**B**) Distribution IHC tumor proportion scores (TPS) and RNA-seq ranks by clinical reporting cutoffs. (**C**,**D**) Median IHC TPS scores by RNA-seq rank clinical reporting cutoff and medial RNA-seq median rank scores by IHC TPS clinical reporting cutoff, Kruskal-Wallis test with significance values adjusted by Bonferroni correction for multiple tests (*p* < 0.001). (**E**) The proportion of patients in each IHC clinical reporting cutoff group was significantly associated with the proportion of patients in the corresponding RNA clinical reporting cutoff group (Chi-Square < 0.001). (**F**) Photomicrographs depict representative PD-L1 IHC staining for each of the subgroups defined as high, low or negative TPS scores within each of the RNA percentile rank clinical reporting categories (high or not high). Scale bars are 100 μm.

**Figure 2 cancers-15-04789-f002:**
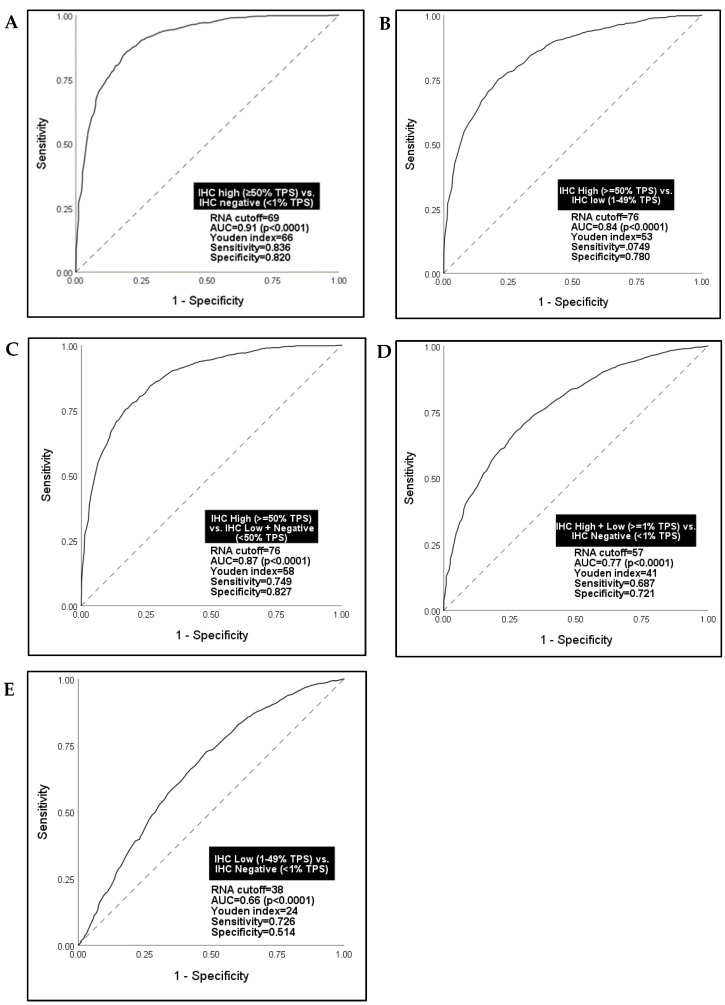
Receiver Operating Characteristic (ROC) analysis for alternate RNA-seq cutoff values for PD-L1 using IHC clinical cutoffs as the reference standard, NSCLC concordance cohort (n = 3168). (**A**) IHC high vs. IHC negative; (**B**) IHC High vs. IHC low; (**C**) IHC High vs. IHC Low + IHC Negative; (**D**) IHC High + IHC Low vs. IHC Negative, and; (**E**) IHC Low vs. IHC Negative.

**Figure 3 cancers-15-04789-f003:**
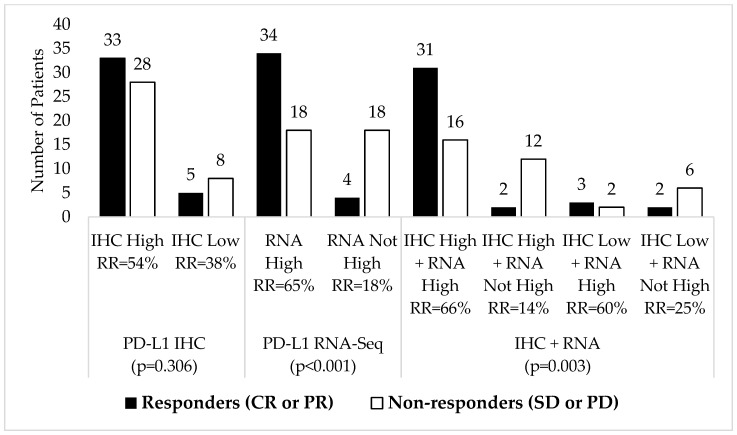
Pembrolizumab response rates (RR) by PD-L1 marker status (n = 74).

**Figure 4 cancers-15-04789-f004:**
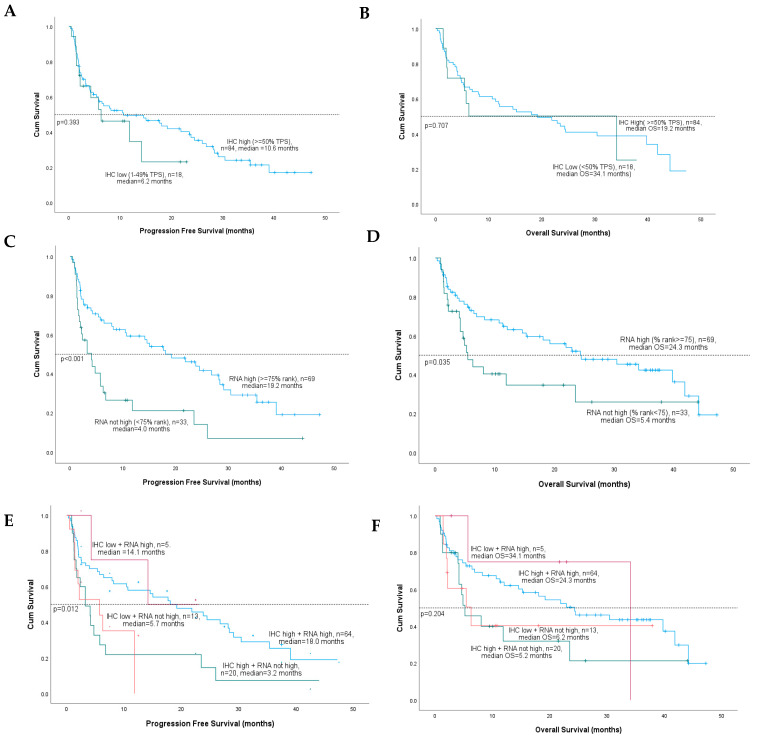
Kaplan-Meier survival curves for median progression free survival (PFS) and median overall survival (OS) by individual or combined PD-L1 IHC and RNA-seq marker status. (**A**,**B**) IHC high vs IHC low; (**C**,**D**) RNA high vs RNA not high, and; (**E**,**F**) IHC + RNA-seq combined status.

**Figure 5 cancers-15-04789-f005:**
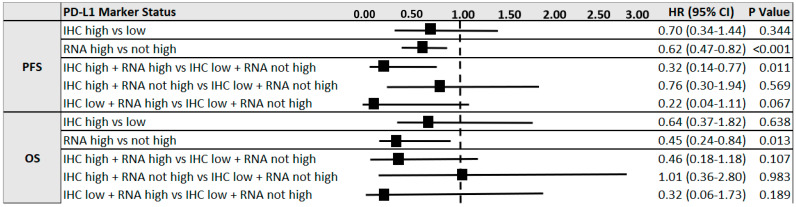
Forest plot of hazard ratios (HR) for predicting progression free survival (PFS) and overall survival (OS) for pembrolizumab by individual and combined PD-L1 marker status for IHC and RNA-seq. Cox proportional regression analyses adjusted for potential covariates: age, sex, histologic subtype, smoking history, weighted performance status, and pembrolizumab line of treatment.

**Table 1 cancers-15-04789-t001:** Pembrolizumab outcomes cohort by PD-L1 RNA-seq expression status (n = 102).

		RNA High (≥75 Rank) n = 69 (67.7)	RNA Not High (<75 Rank) n = 33 (32.3)	Total (n = 102)
PD-L1 IHC status	High (≥50% TPS)	64 (92.8)	20 (60.6)	84 (82.3)
	Low (1–49% TPS)	5 (7.2)	13 (29.4)	18 (17.7)
Age (mean, years)		70	70	70
Sex	Female	39 (56.5)	20 (60.6)	59 (57.8)
	Male	30 (43.5)	13 (39.4)	43 (42.2)
Ever smoker (yes)		61 (88.4)	30 (90.9)	91 (89.2)
Histology	Non-Squamous	61 (88.4)	22 (66.7)	83 (81.4)
	Squamous	8 (11.6)	11 (33.3)	19 (18.6)
Tissue Site	Primary	34 (49.3)	21 (63.8)	55 (53.9)
	Metastatic	35 (50.7)	12 (36.4)	47 (46.1)
Performance status (weighted)	<1	22 (31.9)	3 (9.1)	25 (24.5)
	1 to <2	39 (56.5)	24 (72.7)	63 (61.8)
	2 to <4	6 (8.7)	5 (15.2)	11 (10.8)
Brain metastases (yes)		11 (15.9)	4 (12.1)	15 (14.7)
Pembrolizumab line of treatment	1	59 (85.5)	27 (81.8)	86 (84.3)
	2	8 (11.6)	4 (12.1)	12 (11.8)
	≥3	2 (2.9)	2 (6.1)	4 (3.9)
Prior treatment history (yes)	Chemotherapy	8 (11.6)	5 (15.2)	13 (12.7)
	Targeted therapy	1 (1.4)	2 (6.0)	3 (2.9)
	Immunotherapy	2 (2.8)	0 (0.0)	2(2.0)

## Data Availability

The data presented in this study are available on request from the corresponding author.

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
