# Peer review of "PD-L1 Expression by RNA-Sequencing in Non-Small Cell Lung Cancer: Concordance with Immunohistochemistry and Associations with Pembrolizumab Treatment Outcomes"

_cancers, 2023, doi:10.3390/cancers15194789_

Round 1

Reviewer 1 Report

Reviewer’s Comments

Immune checkpoints are inhibitory regulatory signaling mechanisms operating in immune cells whereby the binding of ligands such as, for example, PD-L1 to their receptors (such as PD-1) leads to the generation in the cells, of intracellular signaling that will inhibit the activation of the cell, for example, through tyrosine phosphatase activation. The pharmacological inhibition of such inhibitory signals leads to enhanced lymphocyte activity. This inhibition can be accomplished by monoclonal antibodies directed, for example, against PD1 or PD-L1 that disrupt their interaction. Importantly, the inhibition of immune checkpoints can enhance antitumor immune responses that can lead in several tumor types including non-small cell lung carcinoma, to clinically meaningful antitumor immune responses, not achievable by other currently used types of treatment. Response to pembrolizumab, an anti-PD-1 antibody is, however, not uniform, and currently the selection of patients for treatment is based on PD-L1 immunohistochemistry (IHC) of tumors. Overall, higher PD-L1 staining is associated with better response, however, the interpretation of low or negative PD-L1 IHC staining intensity can be problematic when used in the context of clinical decision-making due to the relatively low sensitivity of the current IHC protocol.

In the present work Authors compared PD-L1 IHC to PD-L1 quantification by mRNA sequencing in a large set of NSCLC cases and investigated concordance, superiority, and predictive power for response to pembrolizumab treatment.

The reliable selection of patients who will respond to immune checkpoint inhibition therapy is an important goal in the clinic, and is an active research topic in tumor immunology. Given the limitations of the currently used IHC-based PD-L1 detection technique, the present work is important and timely, and will interest a wide range of readers.

Comments:

As stated in lanes 81-82: “NGS cannot distinguish between tumor and immune cell RNA”. Because, as also stated by Authors, PD-L1 expression can be heterogeneous between different regions of a tumor, and because PD-L1 can be expressed not only by tumor cells, but also by cells of the tumor immune infiltrate, quantification of bulk PD-L1 mRNA levels in a tumor can give identical results in rather different situations (such as, for example, homogeneously weakly PD-L1 positive vs. PD-L1 negative and locally highly positive tumors, or PD-L1 expression in tumor cells vs. expression in stromal/immune cells). Incapacity of assigning expression levels to specific cell types/populations is a strong limiting factor when replacing immunohistochemistry by NGS-based approaches is intended without single cell resolution. It would be nice, if Authors could discuss in greater detail, how can it be thought that such an assay that does not discriminate between the tumor and stromal component can be superior or complementary to IHC.

How NGS for PD-L1 can be integrated into the current testing system? Should this replace IHC or should it be performed in conjunction with IHC? Is there a specific IHC category (such as low expression levels by IHC) that should be preferentially tested? What would be the potential consequences of the eventual advent of a more sensitive, higher performance IHC test?

Lanes 43-49: It would be nice, if Authors could state more clearly here the target and the consequent mechanism of action of pembrolizumab in NSCLC. A very brief clear discussion of the core mechanisms of antitumor therapy by immune checkpoint inhibition would probably be useful for non-expert readers. Because pembrolizumab recognizes PD1, while the Paper deals with PD-L1 detection, the specific roles and interactions of these proteins, and the key underlying pharmacological mechanism (pharmacological inhibition of a biological immune-inhibitory signal via antibodies directed either toward the inhibitory ligand or its receptor) is not an automatically intuitive concept that needs to be clearly explained, because some readers who may be experts in NGS may not be fully familiar with the pharmacology of immune checkpoint inhibitors. Probably it would be worthwhile to mention nivolumab as well.

Lanes 89-90 and 437-440, 441-447: ”Western Institutional Review Board (IRB) Copernicus Group”: Please note that the institutional affiliation of this Board/Group may not be clear for all readers. Moreover, it would be nice, if issues (such as anonymity) related to the fact that patient consent was waived could be briefly mentioned.

Please explain, why “This research could not be practicably conducted without a waiver of informed consent … for prospective data collected during standard care that does not yet exist at the time this study is initiated.”

It would be nice if the NGS technologies employed in this work could be presented in a more detailed fashion.

Lanes 98-102: “The pembrolizumab outcomes cohort included patients with stage IIIB/IV disease, who were confirmed negative for ALK and EGFR mutations, received at least one dose of pembrolizumab monotherapy post-CIP test, and had at least 90 days of follow-up (excluding death) post-first dose.” It would be nice, if Authors could discuss here, that this cohort was biased for higher PD-L1 expressing tumors (see also lanes 220, 305-308).

There may be bona fide, biologically real differences in protein vs. mRNA levels for a given protein, for example due to significantly different protein vs. mRNA stability or transcription intensity, independently of differing sensitivities of protein vs. mRNA detection methods. Maybe this could be briefly discussed (see also lanes 174-179). The possibility of the existence of several distinct biological or detection method-related mechanisms that may underlie the different observed discrepancies in mRNA vs. protein levels could be discussed.

Lanes 110-111: “…percentage of viable tumor cells showing % membrane staining at any intensity…” It should be noted that if this score is independent of the intensity of staining (“any intensity”), this may lead to discrepancies when compared to mRNA quantification by NGS.

Lanes 122-124: “RNA-seq results are matched to clinical trials based on inferred drug target associations in patient clinical reports”: This phrase is not clear.

Lane 127: “and anatomic site”: and anatomic site of metastasis

Fig. 4 : Kaplan-Meier

Figure 1, Panel A: Is it appropriate to display negative values (“-5, -10, -15”) on the horizontal axis of this graph?

Lanes 231-234: “…In contrast, we observed no significant differences in response rates in pembrolizumab monotherapy between IHC high (=50% TPS) versus low (<1% TPS) expression (54% vs. 38%, p=.306)” This observation requires some specific discussion here (see lanes 301-308), because this is in conflict with already existing data (see also lane 256) that indicates that higher expression as measured by IHC is associated with better response rates.

Figure 5: The frame at the bottom of the Figure is not visible in the Manuscript received by the Reviewer.

Lanes 276-278: “While PD-L1 expression results in our study that are simultaneously low by IHC and not high by RNA-seq are technically concordant from an analytical interpretation standpoint, they disagree in clinical interpretation as IHC positive and RNA-seq negative.” It is not clear, why “RNA not high” is considered RNA-seq negative. A tumor with PD-L1 mRNA content of >0% but <49% may very well express biologically relevant amounts of PD-L1 protein (even if not clearly detected by current IHC tests). Lane 288: “cases that are also not high (negative) by RNA-seq” : Please explain, why “not high” is considered equivalent to “negative”.

Authors conclude in lanes 312-317 that “Growing evidence substantiates that the use of any single marker for immunotherapy selection is a suboptimal testing strategy, and that determining the likelihood of immunotherapy response requires measuring many more anti-cancer immune cycle response and resistance markers in the tumor microenvironment, including those related to neoantigens, antigen presentation, DNA repair, oncogenic pathways and their mediators.” This is indeed very plausible. It should be noted, however, that the present work deals with a single marker (PD-L1), and no integration of this marker into a wider set of NGS markers is presented. Moreover, it would be nice, if Authors could state more clearly in the Conclusion, in what, and to what extent does their present work contribute to the improvement of the clinical interpretation of PD-L1 status, and whether changes in terms of routine clinical detection and dosage of PD-L1, or further research directions can be proposed based on this work.

Author Response

Dear Reviewer,

The authors are thankful for your thorough review of our study manuscript PD-L1 expression by RNA-Sequencing in non-small cell lung cancer: Concordance with immunohistochemistry and associations with pembrolizumab treatment outcomes. We have made the below changes to address your comments and insights, which have greatly improved the manuscript.

Response to Reviewer 1 Comments

  • As stated in lanes 81-82: “NGS cannot distinguish between tumor and immune cell RNA”. Because, as also stated by Authors, PD-L1 expression can be heterogeneous between different regions of a tumor, and because PD-L1 can be expressed not only by tumor cells, but also by cells of the tumor immune infiltrate, quantification of bulk PD-L1 mRNA levels in a tumor can give identical results in rather different situations (such as, for example, homogeneously weakly PD-L1 positive PD-L1 negative and locally highly positive tumors, or PD-L1 expression in tumor cells vs. expression in stromal/immune cells).

Incapacity of assigning expression levels to specific cell types/populations is a strong limiting factor when replacing immunohistochemistry by NGS-based approaches is intended without single cell resolution.

It would be nice, if Authors could discuss in greater detail, how can it be thought that such an assay that does not discriminate between the tumor and stromal component can be superior or complementary to IHC.

How NGS for PD-L1 can be integrated into the current testing system? Should this replace IHC or should it be performed in conjunction with IHC? Is there a specific IHC category (such as low expression levels by IHC) that should be preferentially tested? What would be the potential consequences of the eventual advent of a more sensitive, higher performance IHC test?

Author response: We expanded the introduction to include evidence of the limited value of discerning tumor from stroma using IHC in NSCLC [lines 104-120]. We clarified our hypothesis that there may be a role for RNA-seq as a complementary PD-L1 measure [lines 139-143]. To support this, we added data from our outcomes analyses for combined IHC and RNA-seq results for PD-L1 that we did not initially include. These findings demonstrate how RNA-seq can help identify patients that are more or less likely to respond [Figure 3, rows 310-319], [Figures 4E and 4F, lines 321-328]. We also assert in conclusions and future directions that the development of a higher performance IHC test for PD-L1 is moot because all biomarkers are susceptible to detection method bias [rows 395-396], and a single marker of response for immunotherapy is insufficient and future testing strategies need to comprehensively assess the tumor immune microenvironment [lines 397-413]. 

  • Lanes 43-49: It would be nice, if Authors could state more clearly here the target and the consequent mechanism of action of pembrolizumab in NSCLC. A very brief clear discussion of the core mechanisms of antitumor therapy by immune checkpoint inhibition would probably be useful for non-expert readers. Because pembrolizumab recognizes PD1, while the Paper deals with PD-L1 detection, the specific roles and interactions of these proteins, and the key underlying pharmacological mechanism (pharmacological inhibition of a biological immune-inhibitory signalvia antibodies directed either toward the inhibitory ligand or its receptor) is not an automatically intuitive concept that needs to be clearly explained, because some readers who may be experts in NGS may not be fully familiar with the pharmacology of immune checkpoint inhibitors. Probably it would be worthwhile to mention nivolumab as well.

Author Response: We added a description of immune checkpoint inhibitors and their mechanism of action to the introduction [lines 46-55].

  • Lanes 89-90 and 437-440, 441-447: ”Western Institutional Review Board (IRB) Copernicus Group”: Please note that the institutional affiliation of this Board/Group may not be clear for all readers. Moreover, it would be nice, if issues (such as anonymity) related to the fact that patient consent was waived could be briefly mentioned.

Please explain, why “This research could not be practicably conducted without a waiver of informed consent … for prospective data collected during standard care that does not yet exist at the time this study is initiated.”

Author Response: We added to Methods that WCG is an independent IRB and provide the WCG website address [lines 146-147]. To eliminate confusion, we restated the patient informed consent statement to read “The WCG IRB determined this study met requirements for a waiver of informed consent under 45 CFR 46 116(f)[2018 Requirements] 45 CFR 46.116(d) and [Pre-2018 Requirements].” [lines 542-543]. The IRB decision was based on the impracticality of establishing research infrastructure in the thousands of community oncology practices across the United States that order reference laboratory tests as Labcorp customers in the course of standard care.

  • It would be nice if the NGS technologies employed in this work could be presented in a more detailed fashion.

Author Response: We included a more detailed description of NGS for immune gene expression by RNA-seq in Methods [lines 172-189].

  • Lanes 98-102: “The pembrolizumab outcomes cohort included patients with stage IIIB/IV disease, who were confirmed negative for ALK and EGFR mutations, received at least one dose of pembrolizumab monotherapy post-CIP test, and had at least 90 days of follow-up (excluding death) post-first dose.” It would be nice, if Authors could discuss here, that this cohort was biased for higher PD-L1 expressing tumors (see also lanes 220, 305-308).

Author Response: We edited Results to reinforce that the treatment cohort is biased for PD-L1 high expression due to guideline recommendations for pembrolizumab monotherapy treatment [lines 277-279], and further highlight it as a limitation in the discussion [lines 380-381].

  • There may be bona fide,biologically real differences in protein  mRNA levels for a given protein, for example due to significantly different protein vs. mRNA stability or transcription intensity, independently of differing sensitivities of protein vs. mRNA detection methods. Maybe this could be briefly discussed (see also lanes 174-179). The possibility of the existence of several distinct biological or detection method-related mechanisms that may underlie the different observed discrepancies in mRNA vs. protein levels could be discussed.

Author Response: We added brief narrative to the Introduction to acknowledge this very real implication [lines 134-138]. We did not discuss in great detail, as we do not have direct observational data, there are no published data to support an opinion that is immediately applicable to this work.

  • Lanes 110-111: “…percentage of viable tumor cells showing % membrane staining at any intensity…” It should be noted that if this score is independent of the intensity of staining (“any intensity”), this may lead to discrepancies when compared to mRNA quantification by NGS.

Author Response: We concur and added this observation to the discussion pertaining to the skewed distribution of TPS results toward 1% staining [lines 359-363].

  • Lanes 122-124: “RNA-seq results are matched to clinical trials based on inferred drug target associations in patient clinical reports”: This phrase is not clear.

Author Response: We clarified that RNA-seq expression results for immune genes that are drug targets in clinical trials are currently reported for research use only [lines 188-189].

  • Lane 127: “and anatomic site”: and anatomic site of metastasis

Author Response: We added “anatomic” [line 192].

  • 4 : Kaplan-Meier

Author Response: Corrected to include hyphen

  • Figure 1, Panel A: Is it appropriate to display negative values (“-5, -10, -15”) on the horizontal axis of this graph?

Author Response: the scale of the X axis of the histogram was changed to remove the negative values. The group labels were also moved beneath the figure for both 1A and 1B.

  • Lanes 231-234: “…In contrast, we observed no significant differences in response rates in pembrolizumab monotherapy between IHC high (=50% TPS) versus low (<1% TPS) expression (54% vs. 38%, p=.306)” This observation requires some specific discussion here (see lanes 301-308), because this is in conflict with already existing data (see also lane 256) that indicates that higher expression as measured by IHC is associated with better response rates. Explain difference in predictive value when comparing therapies in a clinical trial versus predictive value of different levels of expression within an assay

Author Response: We clarified that high expression by IHC trended toward positive associations with outcomes, but was limited by small subgroup sample size in the discussion.

  • Figure 5: The frame at the bottom of the Figure is not visible in the Manuscript received by the Reviewer.

Author Response: The formatting was corrected so that the information now shows.

  • Lanes 276-278: “While PD-L1 expression results in our study that are simultaneously low by IHC and not high by RNA-seq are technically concordant from an analytical interpretation standpoint, they disagree in clinical interpretation as IHC positive and RNA-seq negative.” It is not clear, why “RNA not high” is considered RNA-seq negative. A tumor with PD-L1 mRNA content of >0% but <49% may very well express biologically relevant amounts of PD-L1 protein (even if not clearly detected by current IHC tests). Lane 288: “cases that are also not high (negative) by RNA-seq” : Please explain, why “not high” is considered equivalent to “negative”.

Author Response: In hindsight, we agree that RNA not high is not the same as RNA negative. We removed this language, and instead, expound further in the discussion on the finding that we could not identify an accurate cutoff for RNA-seq moderate versus low expression due to clinical insensitivity of IHC [lines 356-369].

  • Authors conclude in lanes 312-317 that “Growing evidence substantiates that the use of any single marker for immunotherapy selection is a suboptimal testing strategy, and that determining the likelihood of immunotherapy response requires measuring many more anti-cancer immune cycle response and resistance markers in the tumor microenvironment, including those related to neoantigens, antigen presentation, DNA repair, oncogenic pathways and their mediators.” This is indeed very plausible. It should be noted, however, that the present work deals with a single marker (PD-L1), and no integration of this marker into a wider set of NGS markers is presented. Moreover, it would be nice, if Authors could state more clearly in the Conclusion, in what, and to what extent does their present work contribute to the improvement of the clinical interpretation of PD-L1 status, and whether changes in terms of routine clinical detection and dosage of PD-L1, or further research directions can be proposed based on this work.

Author Response: In conclusions, we reiterate that our findings support further study of RNA-seq testing as complementary to current IHC, particularly to improve selection of patients for immunotherapy with low PD-L1 expression. 

Reviewer 2 Report

To Author:

Immune checkpoint inhibitors have improved outcomes over chemotherapy for patients with non-small cell lung cancer (NSCLC). However, the current IHC method for detecting programmed cell death ligand (PD-L1) expression lacks sufficient sensitivity. In this article, the authors detected the expression of PD-L1 by RNA-seq and found that this method is more sensitive. RNA-seq offers an objective PD-L1 measure that could represent an alternative method for patient immunotherapy selection. I considered this research article to be significant. However, I have several suggestions before it can be accepted.

 Comments:

(1) The mutation of EGFR is an important cause of lung cancer and a key factor affecting the prognosis of lung cancer. The authors should not only consider the expression level of PD-L1 when analyzing the therapeutic effect of immune checkpoint inhibitors, but the mutation of EGFR is also an important factor.

(2) The authors should analyze whether these NSCLC patients received immune checkpoint inhibitor therapy alone or received immune checkpoint inhibitor therapy after chemotherapy.

(3) Multiple references are not properly cited, missing page numbers or journal name, such as 4 and 12.

Author Response

Dear Reviewer,

The authors are thankful for your thorough review of our study manuscript PD-L1 expression by RNA-Sequencing in non-small cell lung cancer: Concordance with immunohistochemistry and associations with pembrolizumab treatment outcomes.

Response to Reviewer 2 Comments

 Comments:

  • The mutation of EGFR is an important cause of lung cancer and a key factor affecting the prognosis of lung cancer. The authors should not only consider the expression level of PD-L1 when analyzing the therapeutic effect of immune checkpoint inhibitors, but the mutation of EGFR is also an important factor.

Author response: Per the Methods section, there are no NSCLC patients with EGFR mutations in the pembrolizumab treatment cohort of the study. EGFR mutations are contraindicated on FDA drug labels for PD-1 axis checkpoint inhibition.

  • The authors should analyze whether these NSCLC patients received immune checkpoint inhibitor therapy alone or received immune checkpoint inhibitor therapy after chemotherapy.

Author response: As shown in Table 1, 84.3% of patients in the treatment cohort were treatment naïve and 12.7% of patients received chemotherapy prior to pembrolizumab monotherapy.

  • Multiple references are not properly cited, missing page numbers or journal name, such as 4 and

Author response: Thank you for pointing this out. All references have been checked and error-corrected.

Round 2

Reviewer 1 Report

Authors addressed successfully most issues raised by the Reviewer. A few questions remain however, which are mainly related to newly added, text as follows:

Lane 53: ”When PD-1 on the surface of T cells (a type of immune cell) binds with PD-L1 on the surface of cancer cells…” :  When PD-1 on the surface of T lymphocytes binds with PD-L1 on the surface of cancer cells… (Reviewer believes that it can reasonably be assumed that readers of this Paper will be cognizant that “T cells” are “a type of immune cells”.)

Lane 56: “Checkpoint inhibitors in standard clinical use target either PD-1 or PD-L1…”: quid for example ipilimumab? (“Checkpoint inhibitors in standard clinical use for NSCLC target either PD-1 or PD-L1.”?)

Lanes 59-62: “PD-L1 binds to PD-1 on activated cytotoxic T-cells, leading to downregulation of T-cell immune response and PD-L1 protein expression. Thus, pembrolizumab is particularly effective in targeting NSCLC tumors that express PD-L1.” It is not entirely clear to the Reviewer, what is the pertinence of the downregulation of PD-L1 protein expression by PD-1-PD-L1 binding for the fact that pembrolizumab is particularly effective in targeting NSCLC tumors that express PD-L1.

Lanes 131-134: “However, unlike IHC, standard NGS for mRNA does not discern tumor from stroma, or quantitatively assign PD-L1 expression levels to specific cell types, although the clinical value of making this distinction using current IHC techniques appears to be limited.” Please add bibliographical References.

It would be nice, if in the “PD-L1 expression testing” section (lanes 160-171) Authors could very briefly discuss, how specimens (tissue blocks? slides?) were obtained/collected from the various centers in this study.

Lanes 163-165: “Tumor enrichment and estimation of tumor cell percentage from formalin-fixed paraffin-embedded (FFPE) tissue was performed through microscopic review and cell selection by a board-certified pathologist.” Please specify the quantitative criteria applied to tumor enrichment and estimation of tumor cell percentage.

Lane 194: “pembrolizumab treated patients” : pembrolizumab-treated patients?

Lanes 295-297: “Patients with “double low” (IHC low + RNA not high) status fared better than patients that were high by IHC but not high by RNA-seq (25% vs 14%, respectively).” It would be nice if Authors could propose some tentative mechanistic explanation for this observation. May some IHC high tumors express stable but non-functional PD-L1 protein? Or may these high IHC results correspond to some form of non-specific IHC staining?

It would be nice if the statistical strength of the findings presented in lanes 294-301 could be discussed there.

Lane 372: “we found…”: We found… 

In Discussion or Conclusions please briefly discuss the fact that NGS in bulk does not discriminate between tumor and stromal PD-L1 expression, and whether this may or may not act as a confounding factor in clinical decision-making regarding pembrolizumab therapy.

Lane 91: “investigators concluding PD-L1 expression in NSCLC is tumor-specific.”: Please discuss whether this conclusion obtained using IHC can be extrapolated also to mRNA quantification.  

Lane 130: “ RNA-seq bulk”: “RNA-seq in bulk tissue” or something similar.

Author Response

Dear Reviewer,

Thank you for the additional feedback on our study manuscript PD-L1 expression by RNA-Sequencing in non-small cell lung cancer: Concordance with immunohistochemistry and associations with pembrolizumab treatment outcomes. We have made the below changes to address your comments and insights, which have greatly improved the manuscript.

Response to Reviewer 1 Comments

  • Lane 53: ”When PD-1 on the surface of T cells (a type of immune cell) binds with PD-L1 on the surface of cancer cells…” : When PD-1 on the surface of T lymphocytes binds with PD-L1 on the surface of cancer cells… (Reviewer believes that it can reasonably be assumed that readers of this Paper will be cognizant that “T cells” are “a type of immune cells”.)

 Author response: we removed “a type of immune cell”

  • Lane 56: “Checkpoint inhibitors in standard clinical use target either PD-1 or PD-L1…”: quid for example ipilimumab? (“Checkpoint inhibitors in standard clinical use for NSCLC target either PD-1 or PD-L1.”?)

Author response: we added “NSCLC”

  • Lanes 59-62: “PD-L1 binds to PD-1 on activated cytotoxic T-cells, leading to downregulation of T-cell immune response and PD-L1 protein expression. Thus, pembrolizumab is particularly effective in targeting NSCLC tumors that express PD-L1.” It is not entirely clear to the Reviewer, what is the pertinence of the downregulation of PD-L1 protein expression by PD-1-PD-L1 binding for the fact that pembrolizumab is particularly effective in targeting NSCLC tumors that express PD-L1.

Author response: We agree with the reviewer that it is not pertinent and removed it. We also integrated paragraph 2 into paragraph 1 as it is was somewhat redundant.

  • Lanes 131-134: “However, unlike IHC, standard NGS for mRNA does not discern tumor from stroma, or quantitatively assign PD-L1 expression levels to specific cell types, although the clinical value of making this distinction using current IHC techniques appears to be limited.” Please add bibliographical References.

Author response: We modified this language to more accurately state IHC is variable (and therefore limited) and provide supporting references.

  • It would be nice, if in the “PD-L1 expression testing” section (lanes 160-171) Authors could very briefly discuss, how specimens (tissue blocks? slides?) were obtained/collected from the various centers in this study.

 Author response: We added FFPE specimen requirements

  • Lanes 163-165: “Tumor enrichment and estimation of tumor cell percentage from formalin-fixed paraffin-embedded (FFPE) tissue was performed through microscopic review and cell selection by a board-certified pathologist.” Please specify the quantitative criteria applied to tumor enrichment and estimation of tumor cell percentage.

Author response: We added quantitative criteria (20% tumor purity).

  • Lane 194: “pembrolizumab treated patients” : pembrolizumab-treated patients?

Author response: we added the hyphen to 3 instances of pembrolizumab-treatment where it was missing in the manuscript

  • Lanes 295-297: “Patients with “double low” (IHC low + RNA not high) status fared better than patients that were high by IHC but not high by RNA-seq (25% vs 14%, respectively).” It would be nice if Authors could propose some tentative mechanistic explanation for this observation. May some IHC high tumors express stable but non-functional PD-L1 protein? Or may these high IHC results correspond to some form of non-specific IHC staining?

Author response: a paragraph was added to the discussion (lines 380-387) to address this consideration.

  • It would be nice if the statistical strength of the findings presented in lanes 294-301 could be discussed there.

Author response: We noted that the small numbers for comparisons of combined marker status are a consideration for the strength of our combined marker status findings. (lines 398-399)

  • Lane 372: “we found…”: We found…

Author response: this typo was corrected

  • In Discussion or Conclusions please briefly discuss the fact that NGS in bulk does not discriminate between tumor and stromal PD-L1 expression, and whether this may or may not act as a confounding factor in clinical decision-making regarding pembrolizumab therapy.

Author response: We assert that given tumor or immune cell staining by IHC has been associated with response to checkpoint blockade, that it may not be necessary to discern between tumor and stroma and that NGS sensitivity is more important. [lines 375-379]. We also added a statement to conclusions about computational methods that can be used to determine the contribution of different cell types to overall expression [lines 408-413].

  • Lane 91: “investigators concluding PD-L1 expression in NSCLC is tumor-specific.”: Please discuss whether this conclusion obtained using IHC can be extrapolated also to mRNA quantification.

Author response: this statement is not from the authors of the current manuscript, but rather from the authors of the clinical trial we cited. We believe our comment for item 11 also covers this issue, that discerning tumor from stroma may be less important than assay sensitivity.  

  • Lane 130: “ RNA-seq bulk”: “RNA-seq in bulk tissue” or something similar.

Author response:  we removed this language to avoid confusion as “bulk” was attempting to refer to lack of distinction of tumor/stromal PD-L1 by RNA-seq.